# Elevating Air Temperature May Enhance Future Epidemic Risk of the Plant Pathogen *Phytophthora infestans*

**DOI:** 10.3390/jof8080808

**Published:** 2022-07-30

**Authors:** E-Jiao Wu, Yan-Ping Wang, Li-Na Yang, Mi-Zhen Zhao, Jiasui Zhan

**Affiliations:** 1Institute of Pomology, Jiangsu Academy of Agricultural Sciences/Jiangsu Key Laboratory for Horticultural Crop Genetic Improvement, Nanjing 210014, China; 20190028@jaas.ac.cn (E.-J.W.); njzhaomz@163.com (M.-Z.Z.); 2College of Chemistry and Life Sciences, Chengdu Normal University, Chengdu 611130, China; yanpingwang@cdnu.edu.cn; 3Institute of Oceanography, Minjiang University, Fuzhou 350108, China; linayang@mju.edu.cn; 4Department of Forest Mycology and Plant Pathology, Swedish University of Agricultural Science, 75007 Uppsala, Sweden

**Keywords:** natural selection, AUDPC, ecological sustainability, thermal adaptation, plant disease, pathogen evolution, counter-gradient variation

## Abstract

Knowledge of pathogen adaptation to global warming is important for predicting future disease epidemics and food production in agricultural ecosystems; however, the patterns and mechanisms of such adaptation in many plant pathogens are poorly understood. Here, population genetics combined with physiological assays and common garden experiments were used to analyze the genetics, physiology, and thermal preference of pathogen aggressiveness in an evolutionary context using 140 *Phytophthora infestans* genotypes under five temperature regimes. Pathogens originating from warmer regions were more thermophilic and had a broader thermal niche than those from cooler regions. Phenotypic plasticity contributed ~10-fold more than heritability measured by genetic variance. Further, experimental temperatures altered the expression of genetic variation and the association of pathogen aggressiveness with the local temperature. Increasing experimental temperature enhanced the variation in aggressiveness. At low experimental temperatures, pathogens from warmer places produced less disease than those from cooler places; however, this pattern was reversed at higher experimental temperatures. These results suggest that geographic variation in the thermal preferences of pathogens should be included in modeling future disease epidemics in agricultural ecosystems in response to global warming, and greater attention should be paid to preventing the movement of pathogens from warmer to cooler places.

## 1. Introduction

Public concerns regarding global warming, in terms of societal and ecological sustainability, are increasing. Elevated air temperature and the enhanced intensity and frequency of extreme weather events, such as floods, heatwaves, droughts, and wildfires, which are accompanied by global warming, have already resulted in spatial redistribution and phenological changes in species and loss of biodiversity [1,2,3,4]. According to predictions, if no immediate actions are taken, a 1.5 °C increase in the average global temperature from the preindustrial level may lead to an estimated 50% loss of plant, animal, and insect species [5], ultimately jeopardizing their services to human society and ecosystems [6].

In agricultural ecosystems, global warming can have a profound effect on host–pathogen interactions and, therefore, plant disease epidemics and food production [7,8]. As an important abiotic factor influencing nearly all aspects of biophysical and biochemical activities from molecular to ecosystemic scales, changes in air temperature can strongly affect the epidemic development of plant diseases and food production, either directly by modifying the biological process and distribution of crop plants or indirectly by regulating the survival, reproduction, transmission, and evolution of infectious pathogens [9,10,11,12], presenting new disease management challenges for food security and human health in general.

There is a growing interest in understanding the impact of global warming on future disease epidemics and food production [13,14,15]. Elevated temperatures resulting from global warming can affect the completion [16], population density [17], reproductive model [18], and stress tolerance [19] of both hosts and pathogens. Changes in temperature may also alter the mutation rate [20], migration patterns, and thermal preferences [8,21] of both counterparts. However, given the multidimensional role of temperature in hosts, pathogens, and their interactions, the global warming effect might depend on particular host–pathogen interactions. This hypothesis is primarily motivated by the thermal mismatch theory, which proposes that host and pathogen fitness often peaks at different temperatures in experimental settings [22,23]. Furthermore, pathogen infection involves a cascade of biological and biochemical processes, each regulated by enzymes that vary in their temperature preferences [24]. The differential effects of temperature change on these processes and the development of their functional traits may have additive, multiplicative, or antagonistic effects on epidemics [25]. Indeed, theoretical studies have demonstrated that disease development and severity under global warming may increase in some plant–pathogen systems but decrease or remain unchanged in other systems [9,26]. Thus, there is considerable uncertainty regarding future variations in disease epidemics and food production [27,28].

Pathogen adaptation plays a key role in host–pathogen interactions in shaping plant defense and disease epidemics in ecosystems [29,30]. Plant pathogens can adapt to local temperature conditions through either the gradual accumulation of beneficial mutations [31] or the specific regulation of gene transcription, as expressed by genotype–environment interactions [32]. These local adaptation patterns to temperature have been documented in many species [33,34], including plant pathogens [10,35,36,37,38,39]. For example, in *Zymoseptoria tritici*, isolates originating from warmer regions grow faster in vitro under a higher temperature regime than those from cooler regions, and vice versa [10]. Mariette et al. [38] compared three aggressiveness traits (latent period, lesion growth, and spore production) of *Phytophthora infestans* from different climatic zones in Europe and found genotype-specified thermal adaptation and trade-offs among the components of aggressiveness traits. Although these studies have laid the foundations of the patterns and mechanisms of pathogen adaptation to historical thermal conditions, we generally have insufficient knowledge on how pathogens may respond to the unprecedented temperature events associated with global warming for the robust prediction of future disease epidemics and the development of mitigation strategies to safeguard food security [3,40]. For example, the adaptive potential of pathogens to local air temperature is determined by their genetic variation in populations; however, studies on the impact of global warming on the maintenance and expression of genetic variation in thermal adaptation are scarce. Furthermore, models predicting the response of plant disease epidemics to future temperature events are mainly built on the known thermal niches (maximum, optimum, and minimum temperatures and temperature breadth) of pathogen species, in addition to humidity [3]; however, these niches may evolve in response to shifts in local temperature. Recent studies show that *P**. infestans* intrinsic growth rates from different locations can have a 1–2 °C difference in these thermal niches [41,42]. Currently, information on evolutionary shifts in thermal niches and their associated functional traits is lacking for most pathogens; however, advantages in experimental technologies, such as the application of experimental evolution approaches, will revolutionize the generation of such evolutionary data.

In this study, we used a combination of statistical genetics and common garden experiments to explore the mechanisms and evolutionary processes of thermal adaptation of the aggressiveness of *P. infestans* and its implications for disease epidemics and food security under future climate scenarios. Potato (*Solanum tuberosum* L.) is ranked third globally as a food crop according to total production [43]. Late blight caused by *P. infestans* is the most destructive disease of potato, responsible for the Great Irish Famine in the 1840s, and remains the most devastating plant disease [44], particularly in areas experiencing mild summer temperatures and high humidity. Under conducive climatic conditions (16–22 °C and >97% humidity), the pathogen can affect all parts of the potato plant, including leaves, stems, tubers, and can destroy within 2–4 weeks of infection [45]. This polycyclic pathogen is mainly spread by asexual sporangia that contain infective zoospores [38,46]. Pathogen infection is initiated either indirectly via the discharge of zoospores or directly by the generation of germ tubes from sporangia, producing dark green, brown, and black spots on the surface of potato tissues [47]. A new generation of sporangia can be formed and mature inside potato tissues as quickly as five days [48] and then released from opening stomata through sporangiophores [49]. The annual economic loss caused by late blight is ~USD 8 billion [43,50]. Although varieties with complete resistance to late blight mediated by gene-for-gene interactions are available, they are usually ephemeral. Therefore, most popular varieties are susceptible or partially resistant [51], and fungicide supplementation is needed for disease management.

Aggressiveness, a quantitative component of pathogen pathogenicity in host plants lacking complete resistance [52], is a major ecological trait that determines epidemics. Aggressiveness is a combined measurement of the functional characteristics of the pathogen, including its infection efficiency, sporulation rate, latent period, and lesion growth [53,54]. Unlike virulence profiles (i.e., the ability of a pathogen strain to induce disease or not) which are governed by major genes with simple inheritance [55], aggressiveness is controlled by many genes distributed throughout the pathogen genome, with each gene contributing a minor but additive effect to the phenotypic value of this quantitative trait [56], and is usually more sensitive to environmental variations, such as changes in temperature [57], providing a unique opportunity to study pathogen adaptation to global warming. Therefore, empirical insights into the genetic and physiological determinants of aggressiveness and its thermal preferences and how these genetic, physiological, and ecological characteristics may respond to changes in air temperature are required for robust projections of future plant disease epidemics and food production worldwide.

The overall aim of this study was to determine the impact of global warming on the evolution and epidemics of plant diseases. Briefly, we assayed 140 isolates of the Irish famine pathogen originating from a gradient of thermal environments under five temperature regimes. Pathogen variation in aggressiveness and thermal profile, i.e., maximum temperature (*T*_max_), optimum temperature (*T*_opt_), minimum temperature (*T*_min_), and temperature breadth (*T*_b_) for the infection, was compared to the annual mean temperature at the collection sites and/or experimental temperature. Phenotypic variation in pathogen aggressiveness was also partitioned into values generated by genetic effects (heritability), and genotype–environment integration (plasticity). Neutral variation was estimated from SSR markers. Such synergistic analysis of interactions among genetic variation, evolutionary mechanisms, historical adaptation, experimental temperature and the development of important pathogen traits involving large samples is rare but fundamental to addressing the growing public concerns about human health, food production, and ecological sustainability under global warming. The specific objectives of this study were to (i) determine the differentiation of aggressiveness in the 140 *P. infestans* genotypes collected from seven geographic locations (Appendix A), each representing different temperature zones; (ii) evaluate the impact of local temperature on the development of thermal biology in *P. infestans*; (iii) determine genetic and evolutionary mechanisms driving the thermal adaptation of aggressiveness in *P. infestans*; and (iv) infer the response of *P. infestans* aggressiveness to changes in experimental temperature and the potential impact of global warming on the evolutionary patterns of *P. infestans* and epidemics of late blight disease. We hypothesized that pathogen populations from different geographic regions vary widely in their ability to cause disease, and pathogen populations from warmer regions pose a greater threat to food production by raising future temperatures.

## 2. Materials and Methods

### 2.1. Phytophthora infestans Collection and Isolation

Pathogen isolates were collected from seven fields located in Fujian (Fuzhou and Xiapu), Gansu, Guangxi, Guizhou, Ningxia, and Yunnan along a climatic gradient representing various thermal zones and cropping systems in China during the 2010 and 2011 potato-growing seasons (Appendix A) [41,58]. Guangxi and Fujian have a subtropical climate with hot summers and mild winters, while Gansu and Ningxia have a continental climate with hot summers and cold winters. Potatoes are grown annually in these areas. However, Yunnan and Guizhou are temperate areas where potato can be cultivated all year round. In terms of agro-ecosystems, Guangxi and Fujian are located in the Southern Winter-cropping Region, Yunnan and Guizhou are situated in the Southwest Multi-cropping Region, and Ningxia and northern Gansu are sited in the Northern Single-cropping Region [58]. For each collection, potato leaves with typical late blight symptoms were sampled from different plants separated by at least 100 cm in the early stages of epidemics. Each leaf sample was placed in an individual sandwich bag to prevent cross-contamination. To isolate the pathogen, infected leaves were first rinsed with running water for 60 s and then with sterilized distilled water for 30 s. A segment of a leaf with lesions was placed abaxially on 2.0% water agar for 20–30 h at room temperature in the dark. A piece of mycelium was obtained from a sporulating lesion and inoculated on rye B agar (50 g/L rye and 12 g/L agar) plates supplemented with ampicillin (100 μg/mL) and rifampicin (10 μg/mL). The plates were maintained at 19 °C in the dark for seven days to allow colony development. The isolates were purified three times by sequentially transferring a single sporangium to a fresh rye B plate, generating 96–143 isolates from each location. Only one isolate was obtained from each infected leaf sample. The isolates were genotyped using eight microsatellite markers [59], restriction enzyme-PCR amplification of mitochondrial haplotypes [60], mating type determination [61], and partial sequence analysis of three genes (β-tubulin, Cox1, and Avr3a) [62] and were stored in the dark at 13 °C on rye seeds soaked in sterile water until use. A total of 140 distinct genotypes, with 20 from each of the 7 field populations, were selected for the analysis of thermal-regulated aggressiveness. No Blue-13-A2 lineages were included in the study because the mating type of the selected genotypes was A1 or self-fertile. Details of the pathogen collection, isolation, purification, molecular genotyping, and storage can be found in a previous publication [61,63].

### 2.2. Measurement of the Thermal-Regulated Aggressiveness of Phytophthora infestans

After 4–5 years of storage at 13 °C, the 140 selected isolates were retrieved from long-term storage and transferred twice on fresh rye B medium at 19 °C before the aggressiveness assay. A common garden design was used to test the response of *P. infestans* to different temperatures [37]. Due to strict error control, this experimental design allows us to partition phenotypic variation hierarchically to determine the role of genetic mutation, phenotypic plasticity, and natural selection in the evolution of species adaptation when combined with molecular markers [10,41]. Aggressiveness was tested on detached leaves of Favorite, a potato cultivar susceptible to *P. infestans* [64]. For this test, fully expanded leaves excised from potato plants grown in the field for ~8 weeks were placed on 2% water agar in 9 cm Petri dishes and then infected on the abaxial side with mycelial plugs (*ϕ* = 5 mm) from isolates retrieved from the margins of revived colonies. Petri dishes with detached leaves were exposed to one of five experimental temperatures (13 °C, 15 °C, 19 °C, 22 °C, and 25 °C), and three detached leaves (replicates) were inoculated with each of the 140 isolates for each of the five temperature schemes. These thermal schemes were selected to reflect the ecologically relevant temperatures to which one or more of the pathogen populations were exposed during the growing season (i.e., during the pathogenic phase of its life cycle). Globally, most potato-growing regions fall within this temperature range [65]. The Petri dishes were arranged in a randomized complete block design and maintained at each temperature in an incubator supplemented with 16 h daily light of 30,000 lux.

Lesions formed on the detached leaves were photographed daily from post-inoculation days 2 to 6, and their sizes (areas) were quantified electronically by Assess [66]. Aggressiveness was measured by calculating the area under the disease progress curve (AUDPC) [67] using the estimated lesion sizes. The AUDPC calculated for each isolate at the five temperature regimes was used to generate the thermal reaction norm for aggressiveness. A total of 10,500 lesion data points (140 isolates × 3 replicates × 5 experimental temperatures × 5 days) were included in the evaluation of thermal reaction norms. Owing to the size of the study, the experiment was divided into five sub-experiments, each corresponding to one of the five temperature schemes. To minimize experimental errors, the entire inoculation procedure associated with the process of media preparation, pathogen inoculation, and lesion size estimation at a particular experimental temperature regime was completed by the same person.

### 2.3. Data Analysis

AUDPC, a measure of pathogen aggressiveness, was calculated using the trapezoid integration of lesion sizes over the inoculation period according to the following formula [67]:(1)$$\mathrm{AUDPC}=\overset{n}{\underset{i=1}{\sum }}\left(\frac{x_{i+1}{+x}_{i}}{2}\right){[t}_{i+1}-{\;t}_{i\;}]$$
where x_i+1_ and x_i_ are the lesion sizes at times t_i+1_ and t_i_, and n is the total number of observations. The frequency distribution of AUDPC in the combined population (pool of all isolates from the locations) was visualized by grouping aggressiveness into 11 bins, each differing by 5.00 cm^2^ AUDPC, and tabulated according to each experimental temperature to determine how the experimental temperature affects the frequency distribution of pathogen aggressiveness. The thermal reaction norm of aggressiveness in each isolate was fitted to a quadratic distribution (Y = aT^2^ + bT + c) using the AUDPC (Y) derived from the five experimental temperatures (T). The resulting curves were used to estimate *T*_min_ and *T*_max_ for the aggressiveness of the isolates by setting Y = 0. *T*_opt_ was estimated by taking the derivative of the quadratic equation and solving it in the same way as the calculation of *T_min_* and *T_max_.* The thermal breadth (*T*_b_) for the aggressiveness of an isolate was calculated by taking the difference between its *T*_max_ and *T*_min_. The thermal reaction norm was also used to estimate the maximum aggressiveness of the isolates (i.e., the aggressiveness of the isolates at *T*_opt_). The thermal parameters of pathogens in the populations were correlated with the annual mean temperature of the collection site to determine the impact of local temperature on the formation of pathogen thermal niches.

Variances in aggressiveness were calculated and partitioned into sources attributable to the isolate (*I*, random effect), population (*P*, random effect), and temperature regime (*T*, fixed effect) using the SAS GLM and VARCOMP programs (SAS 9.4 Institute) according to the following model:(2)$$Y_{\mathrm{ript}}=M+I\;\left(P\right)+T+P+I\;\left(P\right){*T+P*T+E}_{\mathrm{ript}}$$
where Y*_ript_* refers to the mean aggressiveness of isolate *i* in replicate *r* for population *p* at temperature *t*, *M* is the overall mean aggressiveness, *T* is the experimental temperature, and *E_ript_* is the variance among the replicates. The terms *P*, *I*(*P*), *I*(*P*)**T*, and *P***T* refer to genetic variance among populations, genetic variance among isolates within populations, variance due to the genotype–environment (temperature) interaction, and different responses of populations to changing temperature, respectively. The population differentiation (*Q*_ST_) in aggressiveness was estimated by calculating the proportion of total quantitative genetic variation due to population variation, as described previously [10,41,63], using the following formula:(3)$${\;Q}_{\mathrm{ST}}=\frac{\delta^{2}{}_{\mathrm{AP}}{+\delta }^{2}{}_{P.E}/n}{\delta^{2}{}_{\mathrm{AP}}{+\delta }^{2}{}_{P.E}{/n+\delta }^{2}{}_{\mathrm{wP}}}$$
where δ^2^_AP_, δ^2^_P_._E_, δ^2^_wp_, and n represent the quantitative genetic variation attributable to among-population differences, variance in population–environment interaction, genetic variation attributable to the difference in isolates within the population, and the number of environments (experimental temperature regimes), respectively. The heritability of aggressiveness in a population was estimated by dividing the genetic variance within populations by the total phenotypic variance [68], while its phenotypic plasticity was calculated by dividing the variance of isolate–temperature interactions by the total phenotypic variance [69]. SSR data for the isolates were obtained from previous studies [70]. Population differentiation in the SSR data was estimated using the fixation index, *F*_ST_ [71]. The standard deviation of *Q*_ST_ was generated from 100 resamples of raw data with replacement and used to test natural selection by comparing *Q*_ST_ and *F*_ST_.

Duncan’s multiple range test [72] was used to compare the aggressiveness, *T*_max_, *T*_opt_, *T*_min_, and *T*_b_ of the *P. infestans* populations. Temperature data for the seven collection sites were obtained from World Climate (http://www.worldclimate.com/ accessed on 10 January 2016). The annual mean temperature at each collection site was estimated from the local temperature recorded over the past 10–20 years, as described previously [41]. To survive in the local area year-round, the pathogen must develop a thermal profile that allows it to infect and multiply not only in the potato host during the pathogenic stage but also during the saprophytic stage when the host is unavailable. Therefore, we believe that the annual mean temperature is a better indicator of the thermal adaptation of the pathogen than the seasonal temperature during the growing months alone. Linear and quadratic models [73] were used to evaluate the associations between parameters, such as pathogen aggressiveness, pathogen thermal niche, temperatures at the collection sites, and experimental temperatures. The interactions of pathogen aggressiveness with annual mean temperature at collection sites and experimental temperatures were visualized using a three-dimensional profile generated using MATLAB 2016a (The Mathworks Inc., Natick, MA, USA).

## 3. Results

### 3.1. Variation and Spatial Differentiation of Aggressiveness in Phytophthora infestans Populations

The aggressiveness of the 140 *P. infestans* isolates displayed wide and continuous variation with a right-skewed mode at the five temperature regimes (Figure 1). Analysis of variance using GLM revealed that “population”, “isolate”, “mean annual temperature of the collection site”, “experimental temperature”, and the interactions between “experimental temperature” and “isolates” significantly contributed (*p* < 0.0001) to the variance of aggressiveness (Appendix A). The average aggressiveness in the seven populations ranged from 8.11–11.30 cm^2^ when the data from the five experimental temperatures were considered together (Table 1, last column). Overall, the pathogens from the warmer locations, for example, Guangxi and Fuzhou, showed higher average aggressiveness than those from cooler places (e.g., Yunnan and Guizhou), and no statistical association was detected between aggressiveness and annual mean temperature at the collection sites when they were evaluated by either the linear (*p* = 0.8027) or quadratic model (*p* = 0.0987).

### 3.2. Impact of Local Temperature on the Thermal Biology of Phytophthora infestans

The estimated *T*_max_, *T*_opt_, *T*_min_, and *T*_b_ of pathogen aggressiveness varied significantly among the populations (Table 2). Unlike the mean aggressiveness (Table 1), the thermal preferences of pathogen aggressiveness were positively associated with the annual mean temperature of the collection sites, although only three (except *T*_min_) of the associations were significant (Figure 2). The *T*_opt_ of aggressiveness in *P. infestans* was ~22 °C, although up to 3.92 °C (Guangxi vs. Ningxia) differences existed among populations sampled from different locations (Table 2). Such among-population differences in thermal preferences also existed for *T*_max_ (up to 6.99 °C) and *T*_b_ (up to 6.89 °C). In contrast, although significant, there was only a small difference among the populations in *T*_min_ (0.03–0.37 °C).

### 3.3. Genetic and Evolutionary Mechanisms Driving the Thermal Adaptation of Aggressiveness in Phytophthora infestans

The plasticity of aggressiveness in the seven populations ranged from 0.41 to 0.73, with an average of 0.55, while heritability in the seven populations ranged from 0.00 to 0.26 with an average of 0.10 (Table 3). The *P. infestans* population collected from Yunnan, a subtropical monsoon climate where potatoes can be planted all year-round, displayed the lowest heritability (0.00), whereas the *P. infestans* population collected from Ningxia, a continental climate with hot summers and cold winters, displayed the highest heritability (0.26). Plasticity in aggressiveness was 2.14–58.96-fold higher than heritability, with an average of 14.02. The overall *Q*_ST_ for aggressiveness in the experiment was 0.17, and natural selection for spatial differentiation was supported by a significantly higher *Q_ST_* than *F_ST_* estimated from SSR markers by a two-tailed *t*-test (*p* < 0.05).

### 3.4. Response of P. infestans Aggressiveness to the Change in Experimental Temperature

The aggressiveness spectrum of the pathogen increased as the experimental temperature increased from 13 °C to 25 °C (Figure 1). The average aggressiveness of the 140 isolates at the five experimental temperatures gradually increased from 0.70 cm^2^ at 13 °C to 18.28 cm^2^ at 22 °C and then decreased to 11.16 cm^2^ as the experimental temperature further increased to 25 °C (Table 1). Model evaluation revealed that the response of *P. infestans* aggressiveness to the change in experimental temperature fitted well with a quadratic distribution (Figure 3A), while the aggressiveness spectrum and experimental temperature had a linear association (Figure 3B). The standard deviation of aggressiveness of the 140 isolates was also positively correlated with experimental temperature (Figure 3C), but the association became insignificant when the standard deviation was normalized by the population mean.

The experimental temperature interacted with the annual mean temperature at the collection sites to determine the aggressiveness of *P. infestans* (Table 1 and Appendix A, Figure 4). At lower-than-expected *T*_opt_, the aggressiveness of *P. infestans* from all geographic locations tended to increase to a similar extent as the experimental temperature increased. On the other hand, at temperatures higher than *T*_opt_, the aggressiveness of *P. infestans* from warmer regions was reduced to a smaller extent than that from cooler regions. For example, the Ningxia population (e.g., from a cooler place) grown at 25 °C was >75% less aggressive compared to that grown at 22 °C. On the other hand, the Guangxi population (e.g., from a warmer place) grown at 25 °C was only ~20% less aggressive compared to that grown at 22 °C. The interactive impact of temperature at collection sites and experiments was also visualized by linear association analysis. At lower experimental temperatures (≤22 °C), pathogen aggressiveness was negatively correlated with the annual mean temperature at the collection sites, suggesting that isolates from warmer places tend to produce less disease (Figure 5). However, the association between the two variables became positive when the experimental temperature increased to 25 °C. Further analysis showed that the correlation coefficients followed a quadratic distribution in response to changes in the experimental temperature (Figure 5).

## 4. Discussion

Global warming has created unprecedented challenges in terms of ecological function and resilience. In agriculture, efforts have been made to predict how such event may affect plant disease epidemics and food production [15]. Although many projections suggest that food security in the future will be markedly jeopardized by global warming [3,74], others have found that the impact could be mild [26,41], depending on particular host–pathogen interactions [14]. However, theoretical forecasts are generally constrained by a lack of understanding of adaptations and trade-offs in the functional traits of pathogens. In this study, we first investigated the patterns and mechanisms of thermal adaptation in pathogen populations and then proceeded to infer how pathogens may adapt to global warming by comparing the functional changes experienced by pathogens with different thermal adaptation histories under a series of experimental temperatures. The local adaptation of pathogen aggressiveness to air temperature is supported by the observed significant difference in AUDPC among *P. infestans* populations sampled from different locations (Table 1), consistent with previous reports on the thermal adaptation of the metabolic rates and other traits of many species [34,75,76], including pathogens [8,77,78]. The higher genetic differentiation in aggressiveness compared to SSR markers suggests that the observed difference is caused by diversifying selection [41,79,80], and local air temperature might be one of the selection agents, as indicated by the strong association between the thermal biology of the pathogen and the annual mean temperature at its collection site (Figure 2).

In many disease projections developed for global changes in air temperature, the constant thermal biology of pathogens and hosts is used. The site-specific thermal preferences observed in many pathogens (e.g., *Phytophthora*
*capsici*, *Pseudomonas syringae*, and *Verticillium dahlia*; *Allophoma tropica*) [35,39] and many other species [33,81] indicate that this treatment in modeling may be suboptimal. *T*_opt_ for mycelial growth and fungicide tolerance in *P. infestans* is ~19 °C [41,42]; however, this temperature for the spore production and aggressiveness development of the pathogen is 10 °C and 22 °C, respectively [38,82,83]. These trade-offs may further complicate the impact of global warming on future disease epidemics and should be considered in future predictions.

The potential for the evolutionary adaptation of ecological traits to environments is determined by genetic variation. Ecological traits with higher genetic variation tend to evolve faster and adapt better to changing environments [84]. Both sequence changes and expression regulation can generate genetic variation [85], are heritable [86], and are responsible for the ecological adaptation of functional traits in which, expression regulation usually initiates the adaptation process and is then enforced by sequence variation [20]. The relative contribution of the two variations to adaptation is dependent on environmental conditions, with a stable environment selecting for permanent adaptation by genetics and a fluctuating environment selecting for temporary adaptation by expression regulation [41,87]. On average, genetic variation measured by heritability accounts for one-tenth of the phenotypic variation in aggressiveness, whereas genotype–environment interaction measured by phenotypic plasticity accounts for more than half of the total variation. This finding suggests that expression regulation plays a more important role than genetic mutations in the development of *P. infestans* aggressiveness in response to temperature change, which is consistent with theoretical expectations and experimental observations of local adaptation to highly fluctuating environmental parameters, such as air temperature [41,42,88]. Apparently, this pattern of the gene mutation–expression relationship was not geography-specific because linear regression analysis did not reveal any association between the plasticity–heritability ratio and local air temperature at the collection sites (*p* = 0.4964). *P. infestans* is capable of long-distance dispersal, and a high degree of plasticity induced by genotype–environment interactions enhances its adaptation when propagules move from one thermal zone to another. Altogether, our results suggest that future global warming may increase pathogen evolution. However, whether this shift in *P. infestans* adaptation will enhance future late blight epidemics depends on the relative adaptation rates of the potato host and pathogen and the trade-offs between aggressiveness and other functional traits of the pathogen. In this study, the annual mean temperature in the seven locations included in the sample ranged from 7.0 to 22.6 °C (Appendix A), and most potato production areas are located in these thermal zones [89]. IPCC [5] predicts that the average air temperature in the next 20 years will increase by ~1.5 °C if there is no concerted effort to reduce greenhouse gas emissions. The 25 °C experimental temperature used in our study represents the expected upper boundary of the future air temperature in the pathogen distribution zones.

Counter-gradient [90] adaptation was observed at low experimental temperatures (Figure 5, left side), suggesting that warmer places may select for *P. infestans* with lower aggressiveness to balance energy allocation [41]. In this situation, *P. infestans* populations from warmer places are expected to cause fewer diseases than those from cooler places. However, the association between *P. infestans* aggressiveness and local temperature at the collection site became positive when the experimental temperature increased to 25 °C (Figure 5, right side). These findings indicate that populations from warmer places might cause a greater threat to plant health and food production than those from cooler places at future elevated air temperatures caused by global warming [41,42,91]. Therefore, future disease quarantine should pay close attention to *P. infestans* from high-temperature areas, such as low latitudes and altitudes [58].

In addition to the increase in average air temperature, global warming is expected to escalate the intensity of spatiotemporal temperature fluctuations, including the frequency of extreme temperature events [5,92]. As *P. infestans* populations originating from warmer places are found to be more thermophilic and demonstrate a broader thermal niche than those originating from cooler places (Figure 2 and Figure 4), *P. infestans* from warmer regions are more aggressive and might be a greater threat to future food production than those from cooler regions [93], further suggesting that future disease quarantine should pay greater attention to pathogens from high-temperature areas.

## 5. Conclusions

Our study revealed that evolutionary adaptation to local environments can generate remarkable within-species variation in the thermal preferences and fitness of *P. infestans*, aligning with the results of other studies on this pathogen [38,41,42] and other species [10,33,34,37]. These results echo public concern that global warming could significantly influence the spatial distribution and severity of many plant diseases [13,94,95]. As the spectrum and intrapopulation variation in pathogen aggressiveness were found to be positively associated with experimental temperature (Figure 3), global warming may increase the evolutionary potential of *P. infestans* aggressiveness and polarize its spatial distribution, increasing the risk of disease epidemics and the challenge of mitigation in future agriculture, which needs urgent attention. *P. infestans* populations from warmer regions are expected to cause more disease than those from cooler regions under future climatic conditions. Therefore, actions should be taken to reduce the movement of pathogens from warmer regions to cooler regions, despite the well-known ability of pathogens to adapt to changing environmental parameters, including temperature [8]. Because we needed to process thousands of inoculations, and common garden designs required us to complete inoculations on the same day, preferably by the same person to minimize errors, we used mycelial plugs instead of sporangia/zoospores in our experiments. This option is not ideal but is not expected to significantly affect our results and conclusions.

## Figures and Tables

**Figure 1 jof-08-00808-f001:**
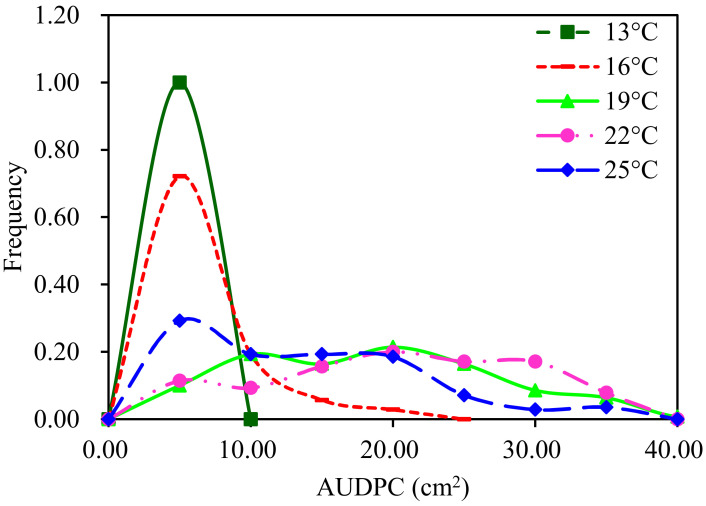
Frequency distribution of aggressiveness in the 140 *Phytophthora infestans* isolates sampled from seven geographic locations with different annual mean temperatures in China. The aggressiveness, represented by AUDPC, was estimated from the lesion sizes of 140 *Phytophthora infestans* isolates at 2–6 days post-inoculation.

**Figure 2 jof-08-00808-f002:**
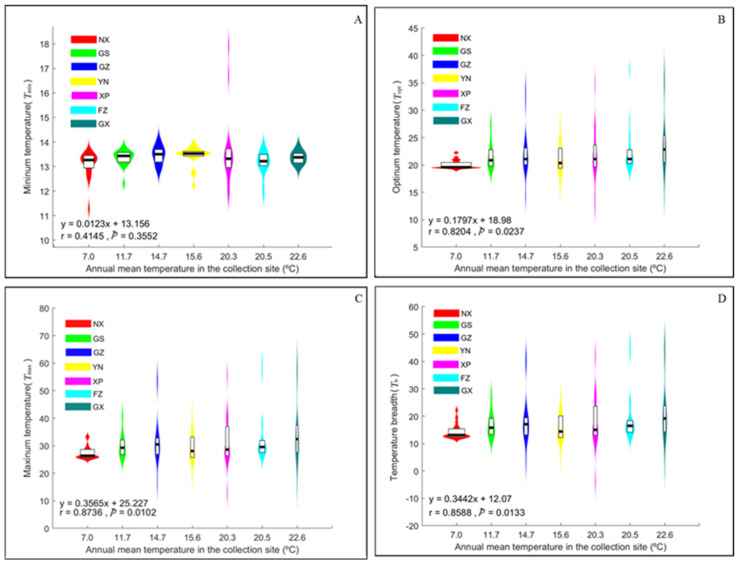
Influence of annual mean temperature at the collection site on the thermal preference of *Phytophthora infestans* aggressiveness. The thermal preference of aggressiveness of each isolate was estimated from the fitted thermal reaction norm. (**A**) Maximum temperature; (**B**) optimum temperature; (**C**) minimum temperature; and (**D**) temperature breadth.

**Figure 3 jof-08-00808-f003:**
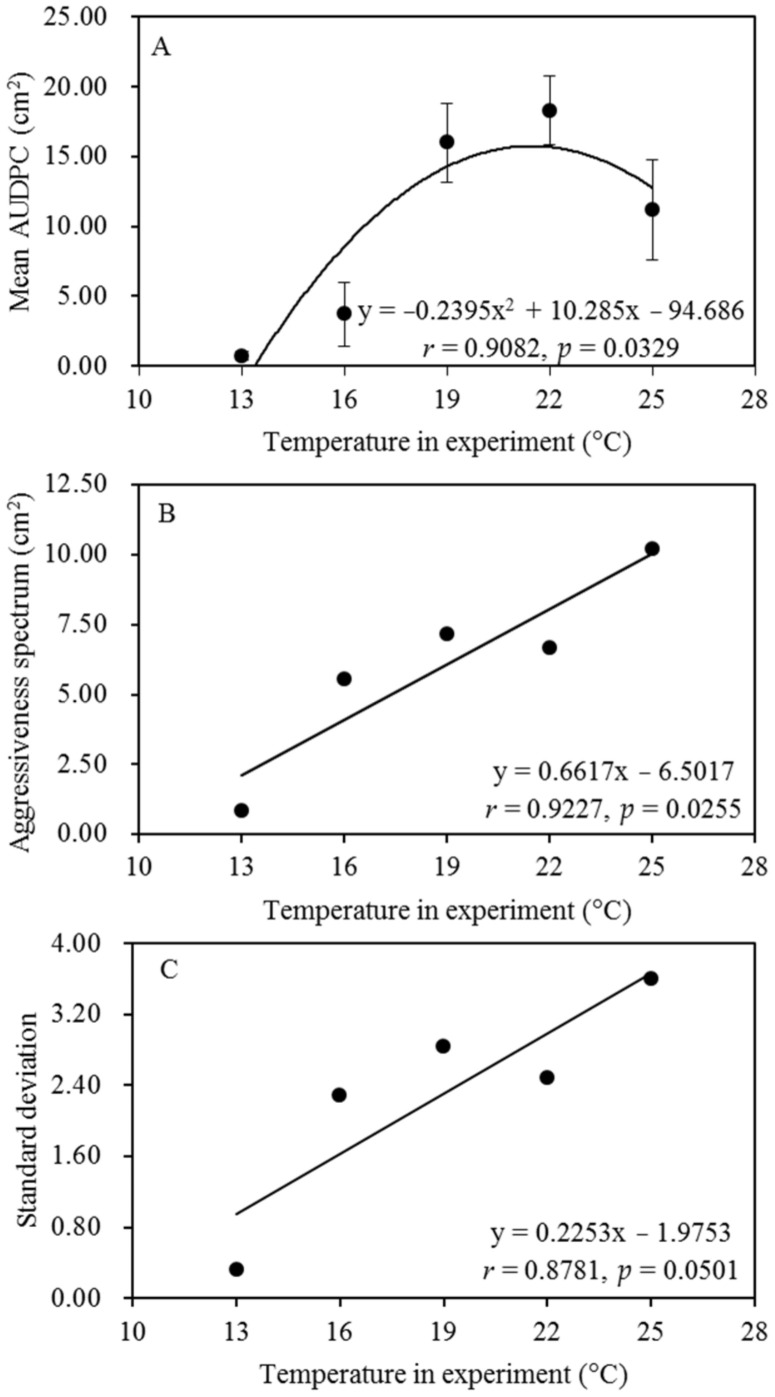
The thermal adaptation pattern of *Phytophthora infestans* aggressiveness estimated from 140 isolates sampled from seven geographic locations with different annual mean temperatures: (**A**) the thermal reaction norm estimated from AUDPC of isolates at the five temperature regimes; (**B**) the spectrum estimated from the difference between isolates with the highest and lowest aggressiveness in the populations; (**C**) standard deviation of aggressiveness of the 140 isolates.

**Figure 4 jof-08-00808-f004:**
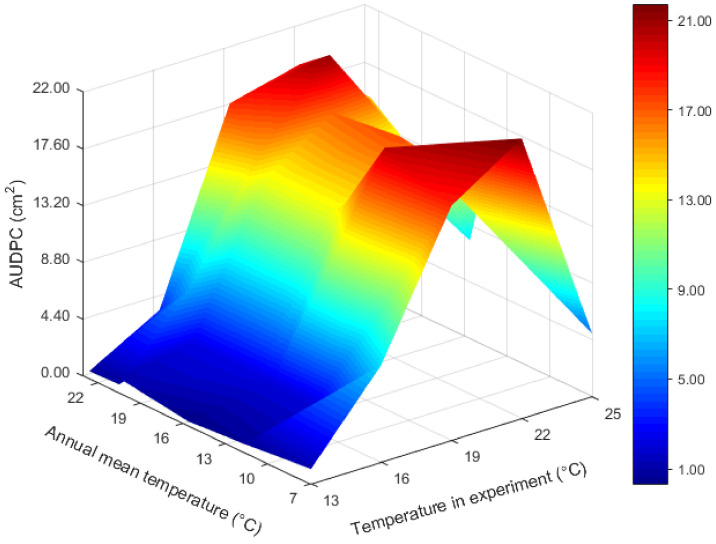
The interactive impact of experimental and historical temperature (i.e., the air temperature at the pathogen collection site) on the development of *Phytophthora infestans* aggressiveness. Pathogen aggressiveness was quantified from 140 isolates sampled from seven geographic locations in China. The aggressiveness, measured by AUDPC, was estimated from lesion size at 2–6 days post-inoculation at five experimental temperatures. NOTE: The bar color shows the change in aggressiveness value from low (blue) to high (red).

**Figure 5 jof-08-00808-f005:**
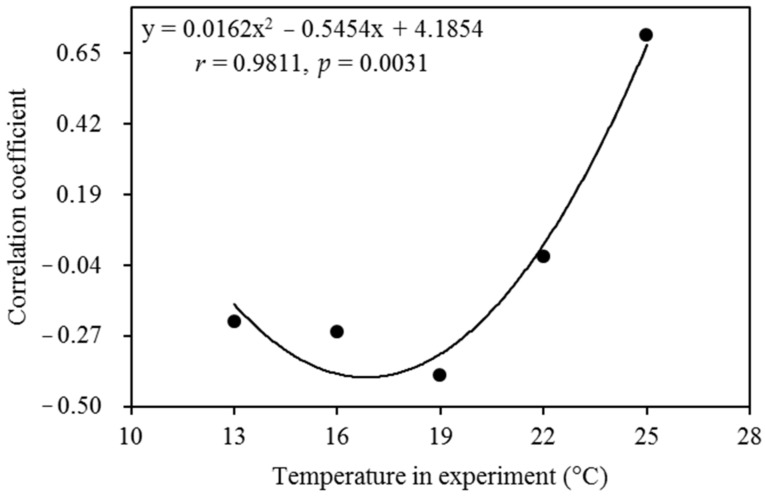
Impact of experimental temperature on the thermal adaptation of *P**hytophthora infestans.* Thermal adaptation was quantified by analyzing the linear association (correlation coefficient) between the aggressiveness of *Phytophthora infestans* and the annual mean temperature at its collection site.

**Table 1 jof-08-00808-t001:** Duncan’s multiple range test for differences in aggressiveness among the seven *Phytophthora infestans* populations. Aggressiveness, represented by AUDPC, was estimated from lesion sizes on potato leaves generated by isolates at 2–6 days post-inoculation.

Population	AUDPC (cm^2^)					
	13 °C	16 °C	19 °C	22 °C	25 °C	Mean
Fuzhou	0.66 ^B^	6.28 ^B^	15.17 ^C^	21.14 ^A^	13.27 ^B^	11.30 ^A^
Xiapu	1.11 ^A^	2.23 ^D^	13.07 ^D^	16.78 ^BC^	11.74 ^B^	8.98 ^C^
Guangxi	0.60 ^B^	3.64 ^C^	17.98 ^B^	19.31 ^AB^	15.07 ^A^	11.32 ^A^
Yunnan	0.33 ^C^	2.14 ^D^	13.08 ^D^	17.39 ^BC^	7.62 ^C^	8.11 ^D^
Gansu	0.64 ^B^	2.06 ^D^	20.23 ^A^	15.03 ^C^	13.32 ^B^	10.25 ^B^
Ningxia	1.15 ^A^	7.52 ^A^	18.18 ^B^	21.69 ^A^	4.89 ^D^	10.69 ^AB^
Guizhou	0.41 ^C^	1.99 ^D^	14.04 ^CD^	16.65 ^B^	12.24 ^B^	9.07 ^C^
Average	0.70	3.69	15.96	18.28	11.16	

Note: Values followed by different letters in the same column are significantly different at *p* = 0.05.

**Table 2 jof-08-00808-t002:** Duncan’s multiple range test for differences in thermal preference among seven *Phytophthora infestans* populations derived from different temperature zones in China. The maximum (*T*_max_), optimum (*T*_opt_), minimum (*T*_min_) temperatures (°C), and temperature breadth (*T*_b_) of aggressiveness were estimated from the thermal reaction norm of each pathogen isolate.

Population	*T* _max_	*T* _opt_	*T* _min_	*T* _b_
Fuzhou	32.62 ^A^	22.45 ^B^	13.17 ^B^	19.45 ^AB^
Xiapu	30.12 ^CD^	21.76 ^BC^	13.10 ^B^	17.02 ^BC^
Guangxi	34.38 ^A^	23.92 ^A^	13.39 ^AB^	20.99 ^A^
Yunnan	29.31 ^DE^	21.05 ^CD^	13.44 ^AB^	15.87 ^DE^
Gansu	30.33 ^CD^	21.73 ^BC^	13.35 ^AB^	16.98 ^CD^
Ningxia	27.39 ^E^	20.00 ^D^	13.29 ^AB^	14.10 ^E^
Guizhou	31.60 ^C^	22.21 ^BC^	13.47 ^AB^	18.13 ^BC^
Average	30.96	21.87	13.35	17.61

Note: Values followed by different letters in the same column are significantly different at *p* = 0.05.

**Table 3 jof-08-00808-t003:** Phenotypic variance, heritability (H), and plasticity (P) of aggressiveness in the *Phytophthora infestans* populations estimated from lesion size at 2–6 days post-inoculation.

Population	Phenotypic Variance	P	H	P:H
Fuzhou	61.48	0.54	0.18	2.92
Xiapu	45.70	0.73	0.01	58.96
Guangxi	72.02	0.55	0.09	6.41
Yunnan	50.21	0.41	0.00	-
Gansu	70.39	0.62	0.07	8.93
Ningxia	76.01	0.56	0.26	2.14
Guizhou	52.29	0.44	0.09	4.75
Average	61.16	0.55	0.10	14.02

## Data Availability

The data presented in this study are available in the article and Appendix A.

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
