# Peer review of "Elevating Air Temperature May Enhance Future Epidemic Risk of the Plant Pathogen Phytophthora infestans"

_jof, 2022, doi:10.3390/jof8080808_

Round 1

Reviewer 1 Report

The authors addressed an interesting question.  However, the manuscript is not yet ready for publication. 

The population of P. infestans in China has been in flux for the past several years.  The authors should stipulate how long the genotypes they studied had been in China.  Were these all Blue-13?  Were they different?  That information needs to be included. 

Why was the annual mean temperature used?  What temperatures did the pathogen see during its active phase?  What is the ecology of the pathogen in these locations?  In many parts of the world, the pathogen is active during the growing season, but may be present (in infected tubers) in storage between seasons.  In some places the storage temperature is 4C, in other locations the storage may be higher?  The agro-ecosystem of the diverse locations needs to be described and there needs to be a rationale given for the use of annual mean temperature. 

It is unclear to me if any of the experiments was repeated.  Was each assessment one each genotype done only once?

More information on the collection of isolates needs to be provided in the manuscript.  What clonal lineages were present?  I realize that this population has previously been described, but a summary of that diversity needs to be presented here. 

I’d like to see some of the actual data on temperature effects.  The T-min and the T-max seem different from what I’ve seen in my experience.  It’s well known that zoospores can infect at temperatures below 10 C.  I’ve also found that using agar plugs of mycelium is dramatically less effective than using zoospores in inoculation experiments.

Where are the explicit definitions for T-b, T-min, T-opt, and T-max?  Is T-min the temperature below which there is no disease? 

There were generally a lot of words used in the manuscript, but still important details were missing.  Many of the words were peripheral to the study. 

Specific comments:

L148.  “formation of thermal biology” means nothing to me.  Please describe what it was you did differently.

L185.  Please provide at least a minimal description of a “common garden design” in this manuscript.  I have no idea what a “common garden design” is.

L189.  Inoculation with a mycelial plug is dramatically different from what happens in the field.  In my experience it is much less effective than inoculation with sporangia/zoospores.  Why was the inoculation done in this manner?  Did the isolates from long-term storage not sporulate? If they did not sporulate, which would be my expectation, how valid are the results.  At what temperature were the isolates kept for long-term storage?  At what temperature were the isolates grown prior to inoculation?  What was the length of time between isolate acquisition and experimentation? 

L200-211.  How many times was this experiment done?  It appears to have been done once. 

L229.  Parameters were correlated to annual mean temperature.  Was this also the mean temperature during the time that the pathogen was active? (during the growth of the crop?)

L283-291.  Figure 1.  Table1.  I’m having difficulty understanding Figure 1 and reconciling it with Table 1.  I don’t understand why Figure 1 is not a histogram.  How can there be a frequency of 1.00 for an AUDPC of 5 (?) with zero for 0 and 10 AUDPC at the 13 C temperature.  Why are the dots connected?  A frequency of 1.00 means there were no isolates with an AUDPC other than 5.  Table one clearly shows  other values.  Help? 

L292-3.  I could not find the method for determining the optimal, minimum and maximum temperatures for pathogen aggressiveness.  Are these methods presented somewhere?  If not, they need to be presented.  T-max, T-min and T-opt need to be defined explicitly. 

Figure 2.  There appear to be data points much higher than believable on the figure.  What are these?

Figure 3.  The standard deviation is clearly associated with the mean in 3 C.  Was there a transformation done to eliminate this association? 

Figure 4.  This figure needs to be explained better.  What is the bar on the right side? 

L388.  How can one compare diversity as determined by neutral markers with diversity in aggressiveness?  I did not see where that comparison was actually done?  Isn’t this like comparing apples and oranges?   I would prefer to see some more complete description of the diversity of the isolates as identified by neutral markers.  

L395.  Citations 35 and 39 do not support this assertion. 

L434-435.  This information should go into the Methods section.

L435-440.  This speculation is beyond what should appear in this manuscript.  The authors provide no basis for this assertion.

Author Response

Reviewer 1

  • The population of P. infestans in China has been in flux for the past several years. The authors should stipulate how long the genotypes they studied had been in China.  Were these all Blue-13?  Were they different?  That information needs to be included.

Answer: 1) We chose isolates for the experiment based on GENOTYPE instead of clonal lineage (many genotypes in a single lineage). For P. infestans populations in China, we have identified hundreds of genotypes in >6000 isolates analyzed in the past 10 years (unpublished data). Many of the genotypes appeared once over the years but only a few of them appeared over 5 years. But we are interested in understanding the evolutionary adaptation of P. infestans at the species level rather than individual genotype/clonal lineage level and we achieved this by looking at the average performance of 140 isolates each with different genetic background (ie with different genotypes. 2) No blue-13-A2 was included in the study (L186-187).

  • Why was the annual mean temperature used? What temperatures did the pathogen see during its active phase?  What is the ecology of the pathogen in these locations?  In many parts of the world, the pathogen is active during the growing season, but may be present (in infected tubers) in storage between seasons.  In some places the storage temperature is 4C, in other locations the storage may be higher?  The agro-ecosystem of the diverse locations needs to be described and there needs to be a rationale given for the use of annual mean temperature.

Answer: 1) The justification of using annual mean temperature is presented in L276-281. 2) The agro-ecosystem of the location is presented in L159-168.

  • It is unclear to me if any of the experiments was repeated. Was each assessment one each genotype done only once?

Answer: Yes. The information is in L202-205.

  • More information on the collection of isolates needs to be provided in the manuscript. What clonal lineages were present?  I realize that this population has previously been described, but a summary of that diversity needs to be presented here.

Answer: Provided in L184-189.

  • I’d like to see some of the actual data on temperature effects. The T-min and the T-max seem different from what I’ve seen in my experience.  It’s well known that zoospores can infect at temperatures below 10 C.  I’ve also found that using agar plugs of mycelium is dramatically less effective than using zoospores in inoculation experiments.

Answer: 1) Table 1 shows how temperature affects disease development. 2) First, the data shown in Table 2 is the average over the population and some isolates did produce disease at 10°C. second, we used mycelium not zoospores for the experiment. 3) We agree that sporangia/zoospores are probably better than mycelial plugs but were not feasible in our study given the fact we need to handle thousands of inoculations (140 x 3 x 5) and the experimental design requires us to complete the inoculation at the same day preferably by the same person to minimize error. We have included caveat in the Discussion (L498-502).   

  • Where are the explicit definitions for T-b, T-min, T-opt, and T-max? Is T-min the temperature below which there is no disease?

Answer: Yes, there are defined in L136-138.

  • There were generally a lot of words used in the manuscript, but still important details were missing. Many of the words were peripheral to the study.

Answer: The manuscript was professionally edited by a Wiley language service, and we carefully checked the wording again during the revision.

Specific comments:

  • “formation of thermal biology” means nothing to me.  Please describe what it was you did differently.

Answer: The sentence has been modified (Line 149).

  • Please provide at least a minimal description of a “common garden design” in this manuscript.  I have no idea what a “common garden design” is.

Answer: Described and more citations are added (L194-198).

  • Inoculation with a mycelial plug is dramatically different from what happens in the field.  In my experience it is much less effective than inoculation with sporangia/zoospores.  Why was the inoculation done in this manner?  Did the isolates from long-term storage not sporulate? If they did not sporulate, which would be my expectation, how valid are the results.  At what temperature were the isolates kept for long-term storage?  At what temperature were the isolates grown prior to inoculation?  What was the length of time between isolate acquisition and experimentation?

Answer: 1) We agree that sporangia/zoospores are probably better than mycelial plugs but were not feasible in our study given the fact we need to handle thousands of inoculations (140 x3 x5) and the experimental design requires us to complete the inoculation at the same day preferably by the same person to minimize error. We have included caveat in the Discussion (L498-502). 2) This is another reason why we used mycelial plugs instead of sporangia/zoospores. 3) The isolates were maintained 4-5 years at 13 °C (L192). 4) They were retrieved and transferred twice at 19 °C (optimum temperature) on fresh rye B medium before the aggressiveness assay were started (L193-194).

  • L200-211. How many times was this experiment done?  It appears to have been done once.

Answer: Three. It is presented in L202-205.

  • Parameters were correlated to annual mean temperature.  Was this also the mean temperature during the time that the pathogen was active? (during the growth of the crop?)

Answer: To survive in the local area year-round, the pathogen must develop a thermal profile that allows it to infect and multiply not only in the potato host during the pathogenic stage, but also during the saprophytic stage when the host is unavailable. Therefore, we believe that the annual mean temperature is a better indicator of the thermal adaptation of the pathogen than the seasonal temperature during the growing months alone. We added this argument in the revision. (L498-502)

  • L283-291. Figure 1.    I’m having difficulty understanding Figure 1 and reconciling it with Table 1.  I don’t understand why Figure 1 is not a histogram.  How can there be a frequency of 1.00 for an AUDPC of 5 (?) with zero for 0 and 10 AUDPC at the 13 C temperature.  Why are the dots connected?  A frequency of 1.00 means there were no isolates with an AUDPC other than 5.  Table one clearly shows  other values.  Help?

Answer: The plot was generated using a similar bin system to the histogram. In this system, isolates with an AUDPC difference of less than 5 cm were grouped into the same bin. At 13 °C, all isolates have an AUDPC less than 5 and the bin system plays them together with a frequency of 1.  

  • L292-3. I could not find the method for determining the optimal, minimum and maximum temperatures for pathogen aggressiveness.  Are these methods presented somewhere?  If not, they need to be presented.  T-max, T-min and T-opt need to be defined explicitly.

Answer: The information is added (L235-240).

  • Figure 2. There appear to be data points much higher than believable on the figure.  What are these?

Answer: It is correct. Instead of using the population average, this violin chart shows all data points for 20 isolates (20 data points) in the populations.

  • Figure 3. The standard deviation is clearly associated with the mean in 3 C.  Was there a transformation done to eliminate this association?

Answer: Thanks. This concern is addressed now (L357-358).

  • Figure 4. This figure needs to be explained better.  What is the bar on the right side?

Answer: The detailed information to this figure is replenished, and the right bar is showing the value with the different color.

  • How can one compare diversity as determined by neutral markers with diversity in aggressiveness?  I did not see where that comparison was actually done?  Isn’t this like comparing apples and oranges?   I would prefer to see some more complete description of the diversity of the isolates as identified by neutral markers. 

Answer: 1) We compare population differentiation, not diversity. 2) this result is presented in the last lines of section 3.3 (The overall QST for aggressiveness in the experiment was 0.17 and natural selection for spatial differentiation was supported by a significantly higher QST than FST estimated from SSR markers by a two-tailed t-test) and the method to achieve the comparison is described in Equation 3 and immediately following. 3) This analysis, known as the Qst/Fst comparison, has been widely used to determine whether and in what state of selection a quantitative trait is (Yang et al., 2016, Wu et al., 2019). The supported theory is that genetic drift affects the entire genome equally, whereas population differentiation in neutral genomes (such as SSR) is caused only by random drift. Any statistical differences from neutral expectations, i.e., differentiation indices in SSR markers, are assumed to be due to selection

  • Citations 35 and 39 do not support this assertion.

Answer: Thanks. The references were changed as the listed below.

Yin, J.; Jackson, K. L.; Candole, B. L.; Csinos, A. S.; Langston, D. B.; Ji, P. Aggressiveness and diversity of Phytophthora capsici on vegetable crops in Georgia. Annals of Applied Biology, 2012, 16, 191-200. https://doi.org/10.1111/j.1744-7348.2012.00532.x

Huot, B.; Castroverde, C. D. M.; Velásquez, A. C.; Hubbard, E.; Pulman, J A.; Yao, J.; Childs, K. L.; Tsuda, K.; Montgomery, B. L.; He, S. Y. Dual impact of elevated temperature on plant defence and bacterial virulence in Arabidopsis. Nature Communications, 2017, 8, 1808. https://doi.org/10.1038/s41467-017-01674-2

  • L434-435. This information should go into the Methods section.

Answer: This sentence has been changed as suggested by the reviewer.

  • L435-440. This speculation is beyond what should appear in this manuscript.  The authors provide no basis for this assertion.

Answer: These texts are removed.

Reviewer 2 Report

I congratulate the authors on this paper. They have come up with, and they executed a study, unlike anything I have seen before. I think this was well done. I have made most of my suggestions on the attached document - usually highlighted with a comment.

The one thing I want to see changed is the title. I think this is misleading "Elevating air temperature may enhance the future epidemic risk of an agricultural plant disease."

while the authors say "an agricultural plant disease", I think it should just say Phytophthora infestans.  The should not be concerned about readership; P. infestans remains a hot topic.  They could even include phenotypic plasticity in the title.  I think this is super interes ng and underresearched in Phytophthora, and fungi in general

I have suggested a couple of Phytophthora papers on phenotypic plasticity and a few Phytophthora DMs were climate change has been considered, for growth suitability, certainly not aggressiveness.  However, there are studies on agressiveness with a range of isolates linked to host or geographic location, and I think some of this literature should be cited.  For example

Yin, J., Jackson, K.L., Candole, B.L., Csinos, A.S., Langston, D.B. and Ji, P., 2012. Aggressiveness and diversity of Phytophthora capsici on vegetable crops in Georgia. Annals of Applied Biology, 160(2), pp.191-200.

Author Response

Reviewer 2

  • I congratulate the authors on this paper. They have come up with, and they executed a study, unlike anything I have seen before. I think this was well done. I have made most of my suggestions on the attached document - usually highlighted with a comment.

Answer: Thank you very much for your positive comments.

  • The one thing I want to see changed is the title. I think this is misleading "Elevating air temperature may enhance the future epidemic risk of an agricultural plant disease."while the authors say "an agricultural plant disease", I think it should just say Phytophthora infestans. The should not be concerned about readership; P. infestans remains a hot topic.  They could even include phenotypic plasticity in the title.  I think this is super interes ng and underresearched in Phytophthora, and fungi in general

Answer: The title of this paper has been changed to “Elevating air temperature may enhance future epidemic risk of the plant pathogen Phytophthora infestans.

  • I have suggested a couple of Phytophthora papers on phenotypic plasticity and a few Phytophthora DMs were climate change has been considered, for growth suitability, certainly not aggressiveness. However, there are studies on agressiveness with a range of isolates linked to host or geographic location, and I think some of this literature should be cited.  For example

Yin, J., Jackson, K.L., Candole, B.L., Csinos, A.S., Langston, D.B. and Ji, P., 2012. Aggressiveness and diversity of Phytophthora capsici on vegetable crops in Georgia. Annals of Applied Biology, 160(2), pp.191-200.

Answer: Thanks. These citations are added.

Additionally, all comments made directly by reviewer to the PDF file have been addressed during revision.